# Psychosocial Correlates of Recreational Screen Time among Adolescents

**DOI:** 10.3390/ijerph192416719

**Published:** 2022-12-13

**Authors:** Joanie Roussel-Ouellet, Dominique Beaulieu, Lydi-Anne Vézina-Im, Stéphane Turcotte, Valérie Labbé, Danielle Bouchard

**Affiliations:** 1Département des Sciences de la Santé, Université du Québec à Rimouski (UQAR), Campus de Lévis, 1595 Boulevard Alphonse-Desjardins, Lévis, QC G6V 0A6, Canada; 2Centre de Recherche du CISSS de Chaudière-Appalaches, 143 Rue Wolfe, Lévis, QC G6V 3Z1, Canada; 3Axe Santé des Populations et Pratiques Optimales en Santé, Centre de Recherche du CHU de Québec, 2400 Avenue D’Estimauville, Québec, QC G1E 6W2, Canada; 4CHAU-Hôtel-Dieu de Lévis, 143 Rue Wolfe, Lévis, QC G6V 3Z1, Canada; 5Laboratoire du Sommeil, Hôtel-Dieu de Lévis, 143 Rue Wolfe, Lévis, QC G6V 3Z1, Canada

**Keywords:** screen time, concurrent screen use, adolescent, correlates, Reasoned Action Approach

## Abstract

The study objective was to identify the psychosocial correlates of recreational screen time among adolescents. Data collection took place in four high schools from the Chaudière-Appalaches region (Quebec, Canada) from late April to mid-May 2021. A total of 258 French-speaking adolescents (69.8% between 15 and 16 years and 66.3% girls) answered an online questionnaire based on the Reasoned Action Approach. Recreational screen time was measured using the French version of a validated questionnaire. Adolescents reported a mean of 5 h and 52 min/day of recreational screen time. Recreational screen time was associated with being a boy (β = 0.33; *p* < 0.0001) and intention to limit recreational screen time to a maximum of 2 h/day (β = −0.15; *p* = 0.0001); this model explained 30% of the variance in behavior. Intention to limit recreational screen time to a maximum of 2 h/day in the next month was associated with attitude (β = 0.49; *p* < 0.0001), self-identity (β = 0.33; *p* < 0.0001), being a boy (β = −0.21; *p* = 0.0109), perceived behavioral control (β = 0.18; *p* = 0.0016), and injunctive norm (β = 0.17; *p* < 0.0001); this model explained 70% of the variance in intention. This study identified avenues to design public health interventions aimed at lowering recreational screen time among this population.

## 1. Introduction

Screens have been present in multiple spheres of our society for many years now. They are used for entertainment, information, and work and as communication tools. The number of Quebec (Canada) households connected to the Internet over the past decade increased from 76% to 93% between 2011 and 2021 [1]. In 2021, 97% of Quebec households owned at least one electronic device, including a smartphone (83%), computer (83%), and tablet (57%) [1]. Among youths (6–17 years), the most commonly used electronic devices in decreasing order were smartphones (74%), computers (70%), gaming consoles (67%), and tablets (62%), and 91% of them used more than one electronic device [2]. A 2021 Quebec survey [2] revealed that 61% of young Quebecers (13–17 years) used different electronic devices for surfing on the Internet for more than 10 h/week.

Many health organizations and associations have issued recommendations to regulate the use of screens by young Canadians. For example, the Canadian 24 h movement guidelines [3] recommend that children and adolescents (5–17 years) limit recreational screen time to a maximum of 2 h/day. However, in 2019, only 20.8% of high school students adhered to this public health recommendation [4]. In 2018–2019, the average recreational screen time among youths (5–17 years) was of 3.8 h/day [5].

According to the World Health Organization [6], excessive screen use would have a significant negative impact on public health and have adverse consequences on the physical, psychosocial, and developmental health of youths. The Digital Health Task Force established by the Canadian Paediatric Society recently documented the effects of screen use on school-aged children and adolescents [7]. Among the health risks identified, there are, for example, a decrease in sleep duration and quality, which affects learning, memory, mood, and behavior; an increase in sedentary lifestyle; and a decrease in time spent on physical activities that increases risks of obesity and cardiometabolic diseases. Eye problems, headaches, fatigue, and an increase in road accidents due to texting while driving are other risks associated with excessive screen use. Developmental and psychosocial health risks include anxiety related to body image and eating disorders, depressive feelings, behavioral or emotional disorders, hyperactivity, relationship problems, online risk taking (e.g., talking to strangers, disclosing personal information), eroded family ties due to limited family time, a decrease in certain learning abilities (e.g., attention, memory), lower academic performance, and increased impulsivity due to multitasking.

Lifestyle habits developed during adolescence tend to be maintained throughout life [8]. It is therefore essential to promote healthy lifestyle habits among this population, who is increasingly seeking autonomy [9]. To develop effective interventions, it is important to have a good understanding of the factors that influence healthy lifestyle habits in this population. Several authors also mention the importance of developing interventions targeting the factors that influence most behavior in order to maximize their potential for success [10,11]. The Theory of Planned Behavior (TPB) [12] and its more recent version, the Reasoned Action Approach (RAA) [13], are both particularly useful to predict and explain health behaviors [14,15,16].

According to the RAA [13], behavior is explained by intention, which reflects the level of motivation towards engaging in this behavior, and perceived behavioral control (PBC), which refers to autonomy and capacity to adopt the behavior. Intention, in return, is influenced by the following three variables: attitude (i.e., a subjective analysis of the advantages and disadvantages (cognitive attitude) as well as positive and negative emotions (affective attitude) associated with adopting a behavior), perceived norms (i.e., the individual’s perception of the social pressure to engage in a behavior (injunctive norm) and perception of the prevalence of the behavior in the environment (descriptive norm)), and PBC. Each of those variables is associated with a specific set of beliefs. Attitude is associated with behavioral beliefs, injunctive norm with normative beliefs, and PBC with control beliefs. Behavioral beliefs are the positive or negative consequences that adopting a given behavior could have according to the individual. Normative beliefs are how an individual thinks specific significant others would react (i.e., approve or disapprove) if s/he adopted the behavior. Control beliefs are factors that can facilitate (i.e., facilitating factors) or limit (i.e., barriers) the adoption of the behavior. According to the RAA, external variables, such as sociodemographic variables (e.g., age, gender, and school level), would have no direct effect on intention or behavior, but rather exert their influence through beliefs.

Since the RAA is open to the inclusion of other variables [13], self-identity was included as a potential correlate of intention and behavior. Self-identity refers to the enduring characteristics that people attribute to themselves, as part of their self-concept (i.e., how people perceive themselves) [17]. In a meta-analysis of the TPB, self-identity showed a sample-weighted correlation of 0.47 with intention for various health behaviors [18]. Self-identity explained an additional 6% of the variance in intention after controlling for TPB variables and past behavior [18] and was associated with intention to engage in health behaviors in youths [19,20]. Habit was also included as an additional variable given the potentially addictive nature associated with screen use [21]. Habit refers to a process whereby a certain context (e.g., leisure time) automatically prompts a specific behavior (e.g., screen time) [22]. Figure 1 presents the theoretical framework used in our study.

Several studies have identified correlates of adolescents’ screen time. A systematic review reported that the most common correlates were age, sex, and body mass index; age and education of parents; socioeconomic status; physical activity; quality of the neighborhood; parents’ screen time and rules; and the presence of electronic devices in the bedroom [23]. Very few studies have used the TPB or the RAA to examine psychosocial correlates of recreational screen time among children and adolescents. The TPB was previously used to predict the screen time of youths and their parents as well as the decision of parents, including mothers only, to limit their child’s screen time [24,25,26].

To our knowledge, only one study previously used the RAA to predict adolescents’ (17–19 years) screen time. This study indicated that this behavior was predicted by intention and habit (β = 0.31, *p* < 0.001, and β = 0.22, *p* = 0.016), and intention to use screens was predicted by positive outcome expectancies (β = 0.47, *p* < 0.001) (e.g., wanting to know about what is going on in the world, communicating with friends or relatives, relaxing), negative outcome expectancies (β = −0.24, *p* < 0.001) (e.g., muscle stiffness and upper back pain, tiredness, isolation, worse shape), descriptive norm (β = 0.18, *p* = 0.006), and PBC (β = 0.18, *p* = 0.003), which explained 26% of the variance in behavior and 34% of the variance in intention [27]. However, none of these studies targeted French-speaking adolescents from Quebec (Canada) and tested self-identity as a correlate of behavior and intention in addition to the RAA variables. To fill this gap in the scientific literature, the present study had two objectives: (1) to identify correlates of recreational screen time among adolescents from the Chaudière-Appalaches region (Quebec, Canada) and (2) to identify correlates of intention to limit recreational screen time to a maximum of 2 h/day in the next month and the most important beliefs associated with this intention. These results will be useful to guide the development of public health interventions, adapted to the specific characteristics of this population, to decrease recreational screen time and their associated health problems among adolescents.

## 2. Methods

### 2.1. Population and Data Collection of the Main Study

The study population consisted of adolescents aged between 14 and 18 years from Chaudière-Appalaches, a French-speaking region of the Province of Quebec (Canada). The project was approved by the Research Ethics Committees of the CISSS de Chaudière-Appalaches (2021-853) and the Université du Québec à Rimouski (CER-115-897). Data collection took place in 12 classes of four high schools from late April to mid-May 2021. Schools were selected in order to represent different socioeconomic statuses [28] and levels of rurality (i.e., one advantaged urban school, one disadvantaged rural school, one advantaged rural school, and one disadvantaged urban school). In each school, principals were asked to select a class of third year (14–15 years), fourth year (15–16 years), and fifth year (16–17 years) of high school. Students who agreed to participate in the study completed an online questionnaire of approximately 15–20 min. Three CAD 25 gift cards for a local sports store were drawn among participants in each class to favor participation (36 in total).

### 2.2. Questionnaire Development

The questionnaire used in this study was developed following the approach suggested by the authors of the RAA [13] and the methodology suggested by Gagné and Godin [29]. A qualitative formative research was first conducted between June and December 2019 to identify the modal salient beliefs associated with limiting recreational screen time to a maximum of 2 h/day in the next month. A sample of 30 French-speaking adolescents aged between 14 and 18 years was randomly selected in five different public places in the Chaudière-Appalaches region (Quebec, Canada): high school, shopping mall, movie theater, youth center, and arena. Participants were recruited based on different sociodemographic characteristics (i.e., sex, school level, location) to ensure a variety of participants. Adolescents interested in participating signed a consent form, and a CAD 5 compensation was given at the end. Individual semistructured interviews lasting about 10–15 min were conducted.

At the beginning of the interview, the following definitions of screens and recreational time were given: “The word *screen* refers to any technology that has a screen. It can be a smartphone, tablet, television, computer, portable gaming system (e.g., Nintendo, PlayStation), etc. and it can be for communicating with your friends (texting, chatting, or emailing), watching television shows or movies (including on the Internet, e.g., Netflix), playing video games, or surfing the Internet” and “In your *recreational time* means do not compute your time during school hours or for school work”. Then, participants were asked eight open-ended questions related to limiting recreational screen time to a maximum of 2 h/day about: (1) advantages and disadvantages of adopting the behavior (behavioral cognitive beliefs), (2) positive and negative emotions related to this behavior (behavioral affective beliefs), (3) people who would agree or not with this behavior (normative beliefs), and (4) barriers and facilitating factors (control beliefs). A qualitative content analysis was realized independently by two experts [30,31] to identify the most important beliefs using a 75% cumulative frequency of mention criterion [13].

The results of this analysis were compared and discussed until a final consensus was reached. The beliefs identified in this step were included in the questionnaire of the present study. This questionnaire was pretested by three behavioral science experts and five adolescents representative of the study population. In December 2020, a test–retest study was then conducted to verify the psychometric qualities of the questionnaire. A total of 38 adolescents (14–15 years) similar to the target population completed the online questionnaire at a 2-week interval.

### 2.3. Questionnaire of the Main Study

The final questionnaire included 50 questions in three sections. The first section contained self-reported behavioral measures of adolescents’ recreational screen time and sleep quality. The second section included measures of potential correlates of recreational screen time among adolescents and the most important beliefs identified in the formative research. This section measures the following RAA variables: intention, attitudes, perceived norms, PBC, behavioral beliefs, normative beliefs, control beliefs, and self-identity and habit. Finally, the third section contains measures of sociodemographic data: age, biological sex, gender, and school level.

#### 2.3.1. Recreational Screen Time

Recreational screen time was measured using a French version of the validated Screen Time-Based Sedentary Behavior Questionnaire [32]. This questionnaire had acceptable test–retest reliability with almost all κ-values > 0.70 among 183 adolescents (12–18 years), and it was also validated with objectively measured sedentary time among 2048 adolescents of the same age [32]. The original questionnaire was translated into French by a certified translator. It contains 12 questions that measure television viewing, computer games, console games, using the Internet for study (educational) and nonstudy (recreational) reasons, and studying during week and weekend days. For each type of screen, adolescents had to estimate their screen time per day. The answer choices were none, <30 min, from 30 min to <1 h, between 1 and <2 h, between 2 and <3 h, between 3 and <4 h, and ≥4 h. In the present study, the items on time spent studying and using the Internet for study reasons were not included because the behavior under study was recreational screen time and some high schools use electronic devices (e.g., tablets) as part of their program. Thus, 8 questions were kept in order to measure screen use for nonstudy reasons (i.e., recreational screen time). The following two questions were added to this section of the questionnaire: (1): “How many hours do you estimate you spend watching screens per week, in your recreational time?” and (2) “Do you use multiple screens at the same time (e.g., watching television and surfing on the Internet on your smartphone) (6-point Likert scale: 1 = always, 6 = never)”. These additional questions were used to verify whether measuring time spent using different screens separately could overestimate total recreational screen time and to assess whether it is common for adolescents to use multiple screens simultaneously (i.e., concurrent screen use).

#### 2.3.2. Sleep Quality

Sleep quality was measured to verify its association with recreational screen time. It was measured using a French version of the validated short version of the Adolescent Sleep–Wake Scale [33,34], which was translated by a certified translator. The results on the association between adolescents’ sleep quality and recreational screen time are already published [35].

#### 2.3.3. Intention

Intention was measured with the following three items on 5-point Likert scales (1 = certainly no, 5 = certainly yes): (1) “I have the intention to limit my recreational screen time to a maximum of 2 h/day in the next month”; (2) “I will try to limit my recreational screen time to a maximum of 2 h/day in the next month”; (3) “I will limit my recreational screen time to a maximum of 2 h/day in the next month”.

#### 2.3.4. Attitudes and Behavioral Beliefs

Attitude was measured using four semantic differentiators with 5-point Likert scales. “For me, limiting my recreational screen time to a maximum of 2 h/day in the next month would be: (1) unpleasant/pleasant; (2) stressful/relaxing; (3) harmful/beneficial; and (4) useless/useful”. Eight items were used to measure the behavioral beliefs with 5-point Likert scales (1 = certainly no, 5 = certainly yes): “If I limited my recreational screen time to a maximum of 2 h/day in the next month, it would: (1) decrease my virtual social interactions (texting, social networks, online gaming community, etc.); (2) improve my concentration; (3) allow me to sleep better; (4) allow me to spend more time with my loved ones; (5) decrease my entertainment activities (video games, reading on the screen, etc.); (6) allow me to have more time to do other things (work, hobbies, etc.); (7) allow me to do more physical activity; (8) make me fear of missing out on important information (news on the Web, social networks, etc.)”.

#### 2.3.5. Perceived Norms and Normative Beliefs

Perceived norms (i.e., injunctive norm and descriptive norm) were measured with the following four items: (1) “Most people who are important to me would recommend that I limit my recreational screen time to a maximum of 2 h/day in the next month (1 = certainly no, 5 = certainly yes)”; (2) “Most people whose opinions I respect would agree that I should limit my recreational screen time to a maximum of 2 h/day in the next month (1 = certainly no, 5 = certainly yes)”; (3) “What percentage of students in your school do you think limit their recreational screen time to a maximum of 2 h/day (0–100%)”; and (4) “Of the three students you know best at your school, how many do you think limit their recreational screen time to a maximum of 2 h/day (none to 3)”. Six items assessed the normative beliefs: (1) “my mother; (2) my father; (3) my friends; (4) my boyfriend/girlfriend; (5) my teachers, and (6) other family members would approve or disapprove that I limit my recreational screen time to a maximum of 2 h/day in the next month (1 = strongly disapprove, 5 = strongly approve or do not apply)”.

#### 2.3.6. Perceived Behavioral Control and Control Beliefs

PBC was measured with the following three items: (1) “For me, limiting my recreational screen time to a maximum of 2 h/day in the next month would be (1 = very difficult, 5 = very easy)”; (2) “If I wanted to, I could limit my recreational screen time to a maximum of 2 h/day in the next month (1 = certainly no, 5 = certainly yes)”; and (3) “I feel capable of limiting my recreational screen time to a maximum of 2 h/day in the next month (1 = certainly no, 5 = certainly yes)”. Control beliefs were measured with perceived barriers and facilitating factors. There were three perceived barriers: “I would feel capable of limiting my recreational screen time to a maximum of 2 h/day in the next month even if: (1) the weather outside is bad, (2) I had no time limit, and (3) my screens were accessible at any time”. There were two facilitating factors: “It would be easier for me to limit my recreational screen time to a maximum of 2 h/day in the next month if (1) I had other things to do (e.g., sports, board games, seeing friends, work, etc.) and (2) I set screen time limits for myself and respect them” (1 = certainly no, 5 = certainly yes).

#### 2.3.7. Self-Identity

Self-identity was measured using two items with 5-point Likert scales (1 = certainly no, 5 = certainly yes): (1) “It fits my personality to limit my recreational screen time to a maximum of 2 h/day”, and (2) “I consider myself to be the kind of person who limits his/her recreational screen time to a maximum of 2 h/day”.

#### 2.3.8. Habit

Habit was measured with the 12-item Self-Report Habit Index [21] using 5-point Likert scales (1 = certainly no, 5 = certainly yes): (1) “I use screens recreationally frequently”; (2) “I use screens recreationally automatically”; (3) “I use screens recreationally without having to remember”; (4) “I feel bad when I don’t use screens recreationally”; (5) “I use screens recreationally without having to think about it”; (6) “I would need to make an effort to not use screens recreationally”; (7) “Recreational screen use is part of my daily routine”; (8) “I start using screens recreationally before I even realize I’m doing it”; (9) “I would find it difficult not to use screens recreationally”; (10) “I use screens recreationally without having to think about it”; (11) “I use screens recreationally because it’s me”; and (12) “I have been using screens recreationally for a long time”.

### 2.4. Statistical Analyses

Sociodemographic data and psychosocial variables were described by frequency, means, and standard deviations. To measure recreational screen time, responses were converted into minutes for each type of screen. Time spent using each screen was added and divided by seven to obtain a total recreational screen time per day [32]. The temporal stability of the French version of the Screen Time-Based Sedentary Behavior Questionnaire, the RAA, and added variables (self-identity and habit) was verified in the test–retest study by computing intraclass correlations (ICC) [36] with their 95% confidence intervals (CI). The criteria of Fermanian [37] were used to classify ICC, whereby results between 0 and 0.30 are very bad, 0.31 to 0.50 are mediocre, 0.51 to 0.70 are moderate, 0.71 to 0.90 are good, and >0.91 are very good. The internal consistency of the RAA and added variables was measured in the test–retest and the main study. The criterion of Nunnally [38] was used to classify Cronbach’s [39] alpha coefficients (α), whereby α > 0.70 are considered acceptable. Pearson correlations were used to verify the internal consistency when a variable had only two items.

To identify correlates of limiting recreational screen time to a maximum of 2 h/day in the next month and those of intention to adopt this behavior, first, ICC using linear mixed models were computed to verify whether the school had an impact on recreational screen time and on intention since it is possible that some schools had adolescents with longer or shorter recreational screen time and with different levels of motivation to limit their recreational screen time. It was planned that if the school had an impact on the behavior or on intention, multilevel analyses would be used to control for the effect of the school on these two variables, and if the school had no impact on recreational screen time or on intention to limit recreational screen time to a maximum of 2 h/day, linear regression analyses would be conducted to identify correlates of behavior and intention.

Since the school had no significant impact on recreational screen time or on intention, linear regression analyses were performed in two phases as recommended by the RAA [13]. Phase 1: to identify correlates of behavior, intention and PBC were first inserted into the model to test whether these variables were correlates of behavior. In a second step, variables added to the RAA (self-identity and habit) were included in the model. Finally, sociodemographic variables were added in a third step. Phase 2: to identify correlates of intention, the psychosocial variables of the RAA (attitudes, perceived norms and PBC) were first entered into the model. In a second step, self-identity and habit were added. Finally, sociodemographic variables were included. The percentage of variance explained was reported for each step of the models. Lastly, a final linear regression model was computed to identify the most important beliefs that influence intention to limit recreational screen time to a maximum of 2 h/day. For each significant RAA variables in the final model, a linear regression analysis with a backward selection of the corresponding indirect belief items on intention was performed [40]. R^2^ shows how well the data fit the regression model. All analyses were conducted using SAS, version 9.4 (SAS Institute, Cary, NC, USA) with a significance level of *p* < 0.05.

## 3. Results

### 3.1. Test–Retest Study and Psychometric Qualities of the Questionnaire

In the test–retest study, the French version of the Screen Time-Based Sedentary Behavior Questionnaire showed a moderate to good temporal reliability according to the criteria of Fermanian [37] (television: ICC = 0.80, 95% CI = 0.61–0.89; computer games: ICC = 0.65, 95% CI = 0.33–0.82; console games: ICC = 0.85, 95% CI = 0.71–0.92; Internet for nonstudy reasons: ICC = 0.77, 95% CI = 0.57–0.88). The internal consistency of the RAA and added variables in the retest–retest study was acceptable with Cronbach’s alpha coefficients ranging from 0.84 to 0.92 [38] and significant Pearson correlations (all *p*’s < 0.05), except for descriptive norm (r = 0.22; *p* = 0.1431). A few minor modifications, such as removing one item of descriptive norm, were made following the test–retest study to foster adequate psychometric properties of the questionnaire before its use in the main study.

The questionnaire used in the main study had acceptable internal consistency according to the criterion of Nunnally [38] with Cronbach’s alpha coefficients ranging from 0.79 to 0.91. The two items of self-identity were significantly correlated (r = 0.47, *p* < 0.0001). Only the three items on perceived norms had a low internal consistency with a Cronbach’s alpha coefficient of 0.32. However, the internal consistency of the two items measuring injunctive norm was acceptable (r = 0.42, *p* < 0.0001).

### 3.2. Main Study

#### 3.2.1. Characteristics of Participants

A total of 271 adolescents agreed to participate and answered the online survey. Among those, 13 were removed from the analyses because they had given unlikely answers or completed the questionnaire in less than 8 min. The average time to complete the online survey was 15 min. The final sample (95.2%) consisted of 258 adolescents (66.3% female (biological sex), while 64.3% identified as girls (gender)). A total of 69.8% of adolescents were 15–16 years of age. Descriptive data of the sample can be found in Table 1.

#### 3.2.2. Recreational Screen Time and Its Correlates

According to the Screen Time-Based Sedentary Behavior Questionnaire, adolescents had an average of 5 h and 52 min/day of recreational screen time. The main source of recreational screen time was the Internet for nonstudy reasons (2 h and 59 min ± 1 h and 11 min.), followed by television (1 h and 16 min ± 1 h and 10 min), console games (57 min ± 1 h and 22 min), and computer games (41 min ± 1 h and 17 min). Based on the additional question on recreational screen time in the questionnaire, adolescents estimated that they spent an average total of 28 h/week (equivalent of 4 h/day). Only 5% of adolescents reported never having concurrent screen use. Very few adolescents (4.7%) adhered to the Canadian 24 h movement guideline of a maximum of 2 h/day of recreational screen time. Results of adolescents’ recreational screen time are presented in Table 2.

To identify correlates of recreational screen time, linear regression analyses were computed since the type of school (advantaged vs. disadvantaged, urban vs. rural) had a nonsignificant impact on adolescents’ recreational screen time (ICC = 0.04, *p* = 0.15). Being a boy (*p* < 0.0001) and intention to limit recreational screen time to a maximum of 2 h/day (*p* = 0.0001) were significant correlates (Table 3). These two variables explained 30% of the variance in behavior. Boys had a higher recreational screen time compared with girls (7 h and 34 min/day vs. 5 h/day, *p* < 0.0001). Differences in recreational screen time between boys and girls are detailed elsewhere [35].

#### 3.2.3. Correlates of Intention to Limit Recreational Screen Time

Linear regression analyses were computed to identify correlates of intention to limit recreational screen time, since the type of school (advantaged vs. disadvantaged, urban vs. rural) had no impact on adolescents’ intention to limit recreation screen time to a maximum of 2 h/day (ICC = 0, *p* = 0.73). The mean of intention was 2.6 ± 1.03 on a 5-point Likert scale, which means adolescents had a neutral intention to limit their recreational screen time to a maximum of 2 h/day in the next month. Five variables were significant correlates of the intention to limit recreational screen time: attitude (*p* < 0.0001), self-identity (*p* < 0.0001), being a boy (*p* = 0.0109), PBC (*p* = 0.0016), and injunctive norm (*p* < 0.0001). This model explained 70% of the variance in intention (Table 4). Girls were more motivated to limit their recreational screen time compared with boys (2.71 ± 1.01 vs. 2.27 ± 1.02, *p* = 0.0012).

#### 3.2.4. Beliefs Underlying the Intention to Limit Recreational Screen Time

The final linear regression model identified five underlying beliefs that were the most strongly associated with the intention to limit recreational screen time to a maximum of 2 h/day. This model explained 50% of the variance in intention. There were two behavioral beliefs: “Limiting recreational screen time to a maximum of 2 h/day in the next month would… improve my concentration (*p* = 0.0001) and decrease my virtual social interactions” (*p* = 0.0164). There was one normative belief: “My friends would approve that I limit my recreational screen time to a maximum of 2 h/day in the next month” (*p* < 0.0001). Finally, there was one facilitating factor: “It would be easier for me to limit my recreational screen time to a maximum of 2 h/day in the next month if I set screen time limits for myself and respect them” (*p* = 0.0012), and one barrier: “I would feel capable of limiting my recreational screen time to a maximum of 2 h/day in the next month even if I had no time limit” (*p* < 0.0001) (Table 5).

## 4. Discussion

This study identified psychosocial correlates of recreational screen time among adolescents and highlighted intervention targets that should be prioritized to help adolescents reduce their recreational screen time. Adolescents reported having an average of 5 h and 52 min/day of recreational screen time using the Screen Time-Based Sedentary Behavior Questionnaire. Recreational screen time was higher than the one reported by the Public Health Agency of Canada in 2018–2019, whereby recreational screen time was about 3.8 h/day [5]. Additionally, very few (4.7%) adhered to the Canadian 24 h movement guideline of a maximum of 2 h/day in the present study. This percentage was higher (20.8%) in 2019 [4]. This could be explained by the fact that the study was conducted during the COVID-19 pandemic. Several studies showed that adolescents’ recreational screen time increased during this period all over the world [41,42], including in Canada [2,43,44].

There were discrepancies between the number of hours adolescents estimated spending weekly in recreational screen time with the additional question and recreational screen time measured using the French version of the Screen Time-Based Sedentary Behavior Questionnaire. Self-reported recreational screen time was lower when it was measured using the additional question. It is difficult to ascertain whether the use of the Screen Time-Based Sedentary Behavior Questionnaire is responsible for this possible overestimation of adolescents’ recreational screen time or if adolescents are not fully aware of how much time they spend using screens recreationally, since there are many methodological issues in measuring accurately screen time [45]. A possible avenue for future research would be to use both validated self-reported and objective measures of adolescents’ recreational screen time, since both types of measure are associated with different biases (e.g., social desirability and memory biases for self-reported measures and reactivity for objective measures).

Recreational screen time was predicted by intention and being a boy. Being a boy was the strongest correlate of adolescents’ recreational screen time. Boys reported an average of 2 and a half hours higher than girls of daily recreational screen time. Previous studies also reported that adolescent boys had a higher recreational screen time compared with girls [46,47,48]. Intention to limit recreational screen time to a maximum of 2 h/day was negatively correlated with recreational screen time. A previous study based on the RAA [27] also identified intention as a predictor of adolescents’ (17–19 years) recreational screen time. However, in that study, habit was another significant correlate of behavior, whereas in our study, this variable was not significantly associated with adolescents’ recreational screen time. Additionally, unlike the present study, adolescents’ recreational screen time did not vary based on the socioeconomic status of their school (advantaged vs. disadvantaged), while a previous systematic review reported that adolescents’ recreational screen time varied by socioeconomic status [23,49]. A possible explanation is that in the present study, only the school’s socioeconomic status was measured, not the adolescents’ or their families’.

Intention to limit recreational screen time to a maximum of 2 h/day was predicted, in decreasing order of importance, by attitude, self-identity, being a boy, PBC, and injunctive norm. Similarly, a previous study based on the RAA reported that adolescents’ (17–19 years) intention to use screens was predicted by positive and negative outcome expectancies (a construct similar to attitude), descriptive norm (but not injunctive norm), and PBC [27]. In a previous study conducted among a similar population (13–18 years) from the same region, self-identity was the strongest correlate of their intention to abstain from consuming sugar-sweetened beverages [19]. In the present study, the mean score for intention represented a neutral intention to limit recreational screen time to a maximum of 2 h/day, indicating that adolescents were not motivated to limit their recreational screen time and suggesting the need for motivational interventions. These results suggest that strategies such as reinforcing a positive attitude by highlighting the positive effects of reducing recreational screen time (e.g., to improve concentration), helping adolescents to develop a strong personal identity as someone who limits his/her recreational screen time to a maximum of 2 h/day (i.e., self-identity), and guiding them to gain a better behavioral control, such as overcoming obstacles, should be part of public health interventions aimed at lowering adolescents’ recreational screen time. In addition, boys should be prioritized by interventions aimed at reducing recreational screen time since they had a higher recreational screen time and were less motivated to limit it compared with girls.

Adolescents’ beliefs regarding their recreational screen time were also identified. These beliefs can be used to develop public health interventions aiming to lower adolescents’ recreational screen time. Two behavioral beliefs, one normative belief, one barrier, and one facilitating factor were significantly correlated with adolescents’ recreational screen time. The behavioral beliefs were that adolescents thought that limiting recreational screen time to a maximum of 2 h/day would improve their concentration and decrease their entertainment activities (video games, reading on the screen, etc.). In fact, previous studies have shown that excessive recreational screen time can be detrimental to adolescents’ ability to concentrate given that it often leads to multitasking (e.g., doing homework and checking cell phone simultaneously) [50,51]. The normative belief was that adolescents’ friends would approve if they limited their recreational screen time, which further confirms the strong influence of peers for this behavior among adolescents [52]. This result suggests including peer network education in interventions aimed at limiting recreational screen time since this approach showed positive results in another health behavior program among children and adolescents (9–14 years) [53]. The most important barrier to overcome was the absence of screen time limits. Although some studies showed that parental rules or control on screens has a significant positive impact on children’s screen time [54,55,56], other studies reported that introducing restrictions on screen time may be less effective for adolescents [57]. This could be explained by the fact that adolescents have access to a variety of screens and that if they are restricted to use a certain type of screen, this will lead them to simply use another type of screen [58] or use screens while they are outside of home. Moreover, to set limits and respect them was a facilitating factor identified in our study. A review of screen time interventions in children (1–12 years) reported that strategies such as self-monitoring, stimulus control, and goal setting were commonly used [57]. These strategies were also successfully used in trials among adolescents [59,60]. As they grow up, adolescents gain more autonomy and become increasingly responsible for their health behaviors, and therefore, promoting self-regulation of screen time may be an important strategy to assist them in both implementing and sustaining healthy screen use [58].

### Strengths and Limitations

This study has a few strengths and limitations. To our knowledge, this is the first study conducted in Quebec (Canada) based on a theory that has the capacity to predict intention and adoption of health behaviors, as supported by a previous meta-analysis [14]. Strengths also include using the French version of a validated questionnaire specifically designed for adolescents to measure recreational screen time, the inclusion of only the items on recreational screen time to be consistent with Canadian public health recommendations on screen time, and verifying whether adolescents had concurrent screen use. Another strength is that the reliability of the questionnaire was measured in a test–retest study prior to the main study.

This study also has a few limitations. One of them is that we may have overestimated recreational screen time since most adolescents reported concurrent screen use. There was a difference between the total weekly recreational screen time reported by adolescents using the additional question in the questionnaire and the total recreational screen time estimated using the Screen Time-Based Sedentary Behavior Questionnaire. Another limitation is that the main study used to identify the psychosocial correlates of adolescents’ recreational screen time was conducted during the third wave of the COVID-19 pandemic (April–May 2021) in Quebec (Canada), while the formative research used to develop the content of the questionnaire based on the RAA was performed before the pandemic (June–December 2019). Therefore, the results should be interpreted with caution as it is possible that they are specific to the context of the COVID-19 pandemic, and additional beliefs specific to the pandemic period may not have been measured. However, our results are in line with previous similar studies conducted before the pandemic [19,27]. Another limitation is that this is a cross-sectional study, with intention and behavior measured at the same time. It would have been preferable to use a longitudinal design and measure intention first and behavior subsequently. However, this strategy was selected to simplify data collection given that it occurred during the COVID-19 lockdown. Future research should confirm the present results in longitudinal studies.

## 5. Conclusions

As adolescents had a high recreational screen time and a very small proportion met the Canadian 24 h movement guideline of a maximum of 2 h/day, public health interventions should suggest limiting recreational screen time and concurrent screen use. This study identified factors that influence adolescents’ recreational screen time and also targets that should be prioritized by public health intervention aimed at promoting a healthy use of screens in this population. For example, boys should be prioritized since they reported a higher recreational screen time and were less motivated compared with girls to limit their recreational screen time. Limiting adolescents’ recreational screen time could have many positive effects, such as possibly improving their physical, mental, psychosocial, and developmental health.

## Figures and Tables

**Figure 1 ijerph-19-16719-f001:**
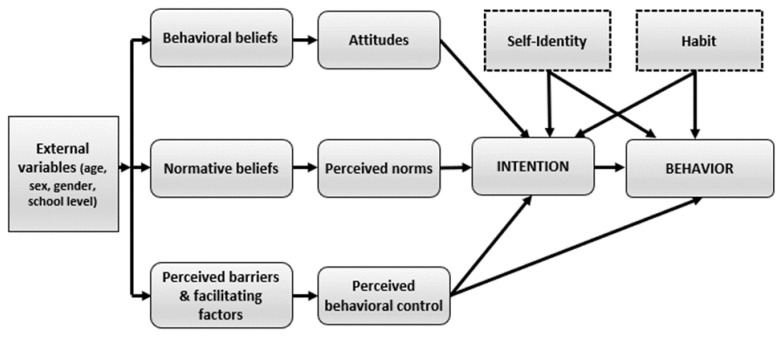
Theoretical framework (adapted from Fishbein and Ajzen, 2010 [13]). Note: Variables in dotted lines were added to the Reasoned Action Approach.

**Table 1 ijerph-19-16719-t001:** Sociodemographic characteristics of participants (*n* = 258).

Variables	*n*	%
*Age*		
14 years	32	12.4
15 years	90	34.9
16 years	90	34.9
17 years	42	16.3
18 years	4	1.5
*Biological sex*		
Female	171	66.3
Male	87	33.7
*Gender*		
Girl	166	64.3
Boy	85	32.9
Neither a girl nor a boy	3	1.2
I prefer not to answer	1	0.4
Other	3	1.2
*School level*		
3rd year of high school	93	36.0
4th year of high school	91	35.3
5th year of high school	74	28.7

**Table 2 ijerph-19-16719-t002:** Adolescents’ recreational screen time (*n* = 258).

Variables	Mean ± SD or %
*Total recreational screen time measured by the Screen Time-Based Sedentary Questionnaire (hours/day)*	5 h and 52 min ± 2 h and 52 min
*Recreational screen time for each screen (hours/day)* Internet for nonstudy reasons	2 h and 59 min ± 1 h and 11 min
Television viewing	1 h and 16 min ± 1 h and 10 min
Console games	57 min ± 1 h and 22 min
Computer games	41 min ± 1 h and 17 min
*Total recreational screen time estimated with the additional question in the questionnaire (hours/day)* *Frequency of concurrent screen use (%)*	4 h ± 3 h
Never	5.0
Rarely	18.6
Sometimes	27.9
Often	27.9
Almost always	17.5
Always	3.1

Note: SD: standard deviation.

**Table 3 ijerph-19-16719-t003:** Correlates of adolescents’ recreational screen time.

Variables	β	SE	*p*-Value *
Intention	−0.15	0.04	**0.0001**
PBC	0.01	0.04	0.7324
Habit	0.02	0.05	0.6349
Self-identity	−0.08	0.04	0.0565
Sex: boy	0.33	0.06	**<0.0001**
Adjusted R^2^: 0.30			

Note: PBC, perceived behavioral control; β, standardized beta; SE, standard error. * Values in bold are statistically significant (*p* < 0.05).

**Table 4 ijerph-19-16719-t004:** Correlates of intention to limit recreational screen time.

Variables	β	SE	*p*-Value *
Attitudes	0.49	0.06	**<0.0001**
Injunctive norm	0.17	0.04	**<0.0001**
Descriptive norm Q1 †	Reference		
Descriptive norm Q2	0.06	0.1	0.5392
Descriptive norm Q3	0.001	0.11	0.9906
Descriptive norm Q4	0.13	0.1	0.2017
PBC	0.18	0.06	**0.0016**
Habit	−0.13	0.07	0.0704
Self-Identity	0.33	0.05	**<0.0001**
Sex: boy	−0.21	0.08	**0.0109**
Adjusted R^2^: 0.70			

Note: PBC, perceived behavioral control; *β*, standardized beta; SE, standard error. * Values in bold are statistically significant (*p* < 0.05). † This variable was divided in quartiles, since it had problematic skewness and kurtosis values; Q1: quartile 1 ≤ 5%, Q2: quartile 2 > 5% and <15%, Q3: quartile 3 ≥ 15% and <25%, Q4: quartile 4 ≥ 25%.

**Table 5 ijerph-19-16719-t005:** Beliefs associated with the intention to limit recreational screen time.

Category	Beliefs	β	SE	*p*-Value *
Behavioral beliefs	*If I limited my recreational screen time to a maximum of 2 h/day in the next month, it would…*			
	(a) Improve my concentration	0.17	0.04	**0.0001**
	(b) Decrease my entertainment activities (video games, reading on the screen, and so on). †	0.10	0.04	**0.0164**
Normative beliefs	My friend *would approve or disapprove if I limited my recreational screen time to a maximum of 2 h/day in the next month.*	0.25	0.05	**<0.0001**
Perceived barriers	*I would feel capable of limiting my recreational screen time to a maximum of 2 h/day in the next month even if* I had no time limit.	0.42	0.04	**<0.0001**
Facilitating factors	*It would be easier for me to limit my recreational screen time to a maximum of 2 h/day in the next month if* I set screen time limits for myself and respect them.	0.16	0.05	**0.0012**
Adjusted R^2^: 0.50				

Note: *β*, standardized beta; SE, standard error. * Values in bold are statistically significant (*p* < 0.05). † Reversed.

## Data Availability

The Research Ethics Committee of the CISSS de Chaudière-Appalaches did not approve publicly sharing our dataset.

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
