# Peer review of "Psychosocial Correlates of Recreational Screen Time among Adolescents"

_ijerph, 2022, doi:10.3390/ijerph192416719_

Round 1

Reviewer 1 Report

Assessment of the quality and innovativeness of the manuscript

The manuscript addresses the relevant question of identification of the psychosocial correlates of recreational screen time among adolescents.

The subject matter is relevant, the scientific quality of the work appears to be sufficient to answer the questions raised.

Knowledge of the relevant state of research and the relevant literature

The authors cite common literature on the subject. A systematic literature review takes place. Comprehensive discussion of the scientific literature throughout the analysis increases the quality of the work.

Appropriateness and accuracy of the theories and methods in addressing the chosen research question

The authors choose relevant statistical methods and approach to address the question.

Stringency of the argumentation

The present study shows consistencies in its argumentative stringency. Discussion section is supported by the empirical data.

Linguistic quality

The linguistic quality of the work is sufficient on the whole. Using too long paragraphs makes it difficult to understand the argumentation in places, paragraphs could be reduced to 6-8 lines.  In some places the authors use a different font.

Other recommendations

In the introduction, the authors write “However, none of these studies targeted French-speaking adolescents” – the absence of studies does not automatically mean the need for them; readers would like to see additional arguments for the coverage of this sample.

In the “Questionnaire Development” section the authors write that “qualitative content analysis was realized independently by two experts” – competence of the experts is not disclosed here.

Author Response

Please see our answers in red italics to each comment of Reviewer 1 and the Assistant Editor.

Reviewer 1

Assessment of the quality and innovativeness of the manuscript

- The manuscript addresses the relevant question of identification of the psychosocial correlates of recreational screen time among adolescents.

The subject matter is relevant, the scientific quality of the work appears to be sufficient to answer the questions raised.

Knowledge of the relevant state of research and the relevant literature

- The authors cite common literature on the subject. A systematic literature review takes place. Comprehensive discussion of the scientific literature throughout the analysis increases the quality of the work.

Appropriateness and accuracy of the theories and methods in addressing the chosen research question

- The authors choose relevant statistical methods and approach to address the question.

Stringency of the argumentation

  • The present study shows consistencies in its argumentative stringency. Discussion section is supported by the empirical data.

We thank the reviewer for the previous positive comments.

Linguistic quality

The linguistic quality of the work is sufficient on the whole. Using too long paragraphs makes it difficult to understand the argumentation in places, paragraphs could be reduced to 6-8 lines.  In some places the authors use a different font.

We have divided some paragraphs that were long to make it easier for readers (see line 90 on page 2, line 157 on page 4, line 171 on page 4, line 305 on page 7, and line 420 on page 10). The article was formatted by the journal. However, we have made some changes to the font on lines 317, 328, 334, and 337 on page 7.

Other recommendations

  • In the introduction, the authors write “However, none of these studies targeted French-speaking adolescents” – the absence of studies does not automatically mean the need for them; readers would like to see additional arguments for the coverage of this sample

It is already mentioned at the end of the introduction that this research will be useful to understand the factors that influence recreational screen time to develop effective interventions in future research (see lines 126-129 on page 3). The Reasoned Action Approach (RAA) is the theory selected because it proved to be particularly useful to predict and explain health behaviors and this is mentioned in the manuscript (see lines 69-71 on page 2). In addition, our study is the only one that targeted French-speaking adolescents. Finally, to our knowledge, it is the first study to test self-identity as a correlate of recreational screen time as an additional variable to the RAA. This information is also already stated in the manuscript (see lines 120-122 on page 3).

  • In the “Questionnaire Development” section the authors write that “qualitative content analysis was realized independently by two experts” – competence of the experts is not disclosed here.

We have added two references of article that used the same methodology conducted by our research team (see references 30 and 31 on line 169 on page 4).

Assistant Editor

I am writing to inform you that the format of the references in the manuscript does not meet the requirements of our journal. Please modify the format according to the attachment.

We modified the references to the ACS format in EndNote.

Reviewer 2 Report

The advantage of this paper is the way of presenting the issues taken under consideration here as well as the reliability of the analyses.  Recreational screen time is a very serious social problem nowadays.  So, it is difficult to overestimate the importance of reliable researches aimed at a comprehensive diagnosis of this phenomenon and its psychosocial correlates. It is also worth emphasizing that the authors decided to focus attention to the practical dimension of the problem. Their intention was also to indicate the factors determining the readiness of adolescents to limit the recreational screen time.

The advantages of this article - its strengths - are primarily:

1.     Good grounding in theory and research conducted so far.

2.     Description of research procedures.

3.     Scope of used methods of empirical data analysis

4.     Method of presenting research results which encourages critical reflection on effective methods of prevention and - what is also important – educational solutions aimed to raise awareness of the negative contexts of recreational screen time.

However, for the sake of the academic soundness of the findings presented in the paper it is worth considering a few detailed issues.

1.      The first is methodological. What is missing here is a precisely formulated research hypothesis justifying the correlations between the variables studied. If, as intended by the authors, the research was of a diagnostic and exploratory nature, then it would be necessary to formulate a list of research problems. The answers to the research questions then determine the order of the narrative during the presentation of the results. The reader then has a chance to follow the logic of the argument.

2.     Another suggestion is editorial. One may wonder whether the presentation of the results with the use of e.g. charts and other graphic materials would facilitate the reception of the content?

3.      In conclusion, it would be worthwhile to include threads indicating the directions of future research and new methodological solutions. I believe that excessive adherence to quantitative research models does not lead to a solution to this problem. It is worth taking this effort - the more so that the mentioned part of the paper is elaborated rather in a laconic way.

Despite the above-mentioned doubts, I state that the research presented in the article meets the criteria of reliability and validity - hence I recommend its publication

Author Response

Please see our answers in red italics to each comment of Reviewer 2 and the Editor.

Reviewer 2

The advantage of this paper is the way of presenting the issues taken under consideration here as well as the reliability of the analyses.  Recreational screen time is a very serious social problem nowadays.  So, it is difficult to overestimate the importance of reliable researches aimed at a comprehensive diagnosis of this phenomenon and its psychosocial correlates. It is also worth emphasizing that the authors decided to focus attention to the practical dimension of the problem. Their intention was also to indicate the factors determining the readiness of adolescents to limit the recreational screen time.

The advantages of this article - its strengths - are primarily:

  1. Good grounding in theory and research conducted so far.
  2. Description of research procedures.
  3. Scope of used methods of empirical data analysis
  4. Method of presenting research results which encourages critical reflection on effective methods of prevention and - what is also important – educational solutions aimed to raise awareness of the negative contexts of recreational screen time.

We thank the reviewer for the positive comments.

However, for the sake of the academic soundness of the findings presented in the paper it is worth considering a few detailed issues.

  1. The first is methodological. What is missing here is a precisely formulated research hypothesis justifying the correlations between the variables studied. If, as intended by the authors, the research was of a diagnostic and exploratory nature, then it would be necessary to formulate a list of research problems. The answers to the research questions then determine the order of the narrative during the presentation of the results. The reader then has a chance to follow the logic of the argument.

The objective of our study is already stated on lines 122-126 on page 3. We did not add research hypotheses, since it would have been redundant with the information contained in the paragraph describing the Reasoned Action Approach (RAA) (see lines 72-89 on page 2). As hypothesized by the RAA, intention and perceived behavioral control (PBC) should be correlates of recreational screen time and attitude, perceived norms, and PBC should be correlates of intention to limit recreational screen time to a maximum of 2 hours/day. Also, our first objective is “to identify correlates of recreational screen time among adolescents from the Chaudière-Appalaches region (Quebec, Canada)” while our second objective is “to identify correlates of intention to limit recreational screen time to a maximum of 2 hours/day in the next month and the most important beliefs associated with this intention”. We used that same order of presentation and logic when we presented our results. We started with a paragraph on recreational screen time (see line 349 on page 8) where we presented the correlates of recreational screen time (see also Table 3 on page 9) and after this, we presented the correlates of intention to limit recreational screen time (see line 374 on page 9 and Table 4 on page 9). Finally, we used the same order of presentation and logic in the discussion. We discussed first the results on recreational screen time (see lines 433-447 on page 11) and after this, the results on intention (see lines 448-466 on page 11).

  1. Another suggestion is editorial. One may wonder whether the presentation of the results with the use of e.g. charts and other graphic materials would facilitate the reception of the content?

The article already contains one figure presenting the theory used in the present study and five tables presenting the results, which we believe is sufficient.

  1. In conclusion, it would be worthwhile to include threads indicating the directions of future research and new methodological solutions. I believe that excessive adherence to quantitative research models does not lead to a solution to this problem. It is worth taking this effort - the more so that the mentioned part of the paper is elaborated rather in a laconic way.

We have added a sentence suggesting that a possible avenue for future research would be to use both validated self-reported and objective measures of adolescents’ recreational screen time, since both types of measure are associated with different biases (e.g., social desirability and memory biases for self-reported measures and reactivity for objective measures) (see lines 428-432 on pages 10 and 11).

Despite the above-mentioned doubts, I state that the research presented in the article meets the criteria of reliability and validity - hence I recommend its publication.

We thank the reviewer for recommending our article for publication.

Editor

Roussel-Ouellet et al. proposed to identify correlates of recreational screen time, and to identify correlates of intention to limit recreational screen time to a maximum of 2 hours/day, among adolescents from the Chaudière-Appalaches region (Quebec, Canada). The research design is sound, and sampling is done adequately to ensure representation for the population of adolescents in the Chaudière-Appalaches region. However, I have a few comments that if addressed, will enhance the quality of the manuscript greatly:

We thank the editor for the positive comments and suggestions to improve the quality of our manuscript.

Major comments:
1. As an exploratory study, I would caution the use scores from unvalidated scales. Computing means of scores from 3 questions on a 5-point Likert scale may appear reasonable. However, the distribution of the scores is likely to dictate the analysis performed. Can the authors confirm if the distribution of scores for the Intention scale was normally distributed? If not, can they show using logistic regression (binary or ordinal) analysis instead of a linear regression?

All scales used in our study were validated (i.e., measure of recreational screen time, sleep quality and also the questionnaire developed for our study). The questionnaire developed for our study was validated in a test-retest study, and it had acceptable temporal stability and internal consistency. The distribution of intention was normal (skewness = 0.3339 and kurtosis = -0.5342). Using a logistic regression analysis instead of a linear regression would result in a loss of statistical power.

  1. The conclusion of the abstract section “This study confirms that adolescents have a high recreational screen time…” is completely inconsistent with the primary objective of the research “The study objective was to identify the psychosocial correlates of recreational screen time among adolescents”. Can the authors please revise this conclusion such that it is appropriately aligned with the objective and principal findings? Further, such a conclusion only sends a confusing picture if the authors highlight that participants’ overestimation of their recreational time use is a limitation.

We agree with the editor and have removed the sentence in the abstract (see line 27 on page 1) and in the discussion (see lines 410-411 on page 10). We thank the editor for pointing out that inconsistency.

Minor comments
Table 1 and 2 - Please take out bulleting of the categories of the variable. Instead, I will suggest using indents.

We removed the bullets in both tables (see Table 1 on page 8 and Table 2 on page 8).